**Data Availability Statement:** All relevant data are within the manuscript.

# *Campylobacter jejuni* and *Campylobacter coli* infection, determinants and antimicrobial resistance patterns among under-five children with diarrhea in Amhara National Regional State, Northwest Ethiopia

**Mesfin Worku**[1]*, **Belay Tessema**[1,2], **Getachew Ferede**[1], **Linnet Ochieng**[3], **Shubisa Abera Leliso**[4], **Florence Mutua**[3], **Arshnee Moodley**[3,5], **Delia Grace**[3,6‡], **Baye Gelaw**[1‡]

1 Department of Medical Microbiology, School of Biomedical and Laboratory Science, College of Medicine and Health Sciences, University of Gondar, Gondar, Ethiopia, 2 Institute of Clinical Immunology, Faculty of Medicine, University of Leipzig, Leipzig, Germany, 3 Animal and Human Health Program, International Livestock Research Institute (ILRI), Nairobi, Kenya, 4 Animal Health Institute, Sebeta, Ethiopia, 5 Department of Veterinary and Animal Science, Faculty of Health and Medical Science, University of Copenhagen, Frederiksberg C, Denmark, 6 Natural Resource Institute, University of Greenwich, London, United Kingdom

‡ DG and BG are contributed equally to this work and share last authorship.
* mesfinwh@gmail.com

## Abstract

### Background

Children with under-five year age disproportionally affected with foodborne illness. Campylobacteriosis is the most common foodborne disease next to Norovirus infection. Macrolides are commonly prescribed as the first line of treatment for Campylobacter gastroenteritis, with fluoroquinolone and tetracycline as secondary options. However, resistance to these alternatives has been reported in various regions worldwide.

### Objective

To determine the prevalence, associated risk-factors and antimicrobial resistance of *Campylobacter jejuni* and *C. coli* among under-five children with diarrhea.

### Methods

Institution-based cross-sectional study was conducted from November, 2022 to April 2023. The study sites were selected using a random sampling technique, while the study subjects were included using a convenient sampling technique. The data were collected using a structured questionnaire. Stool samples were inoculated onto modified charcoal cefoperazone deoxycholate agar and incubated for 48 hours. The suspected colonies were analyzed using matrix-assisted laser desorption ionization-time of flight mass spectrometry to confirm the species. Antimicrobial susceptibility testing was performed using a disc diffusion technique. All potential covariates (independent variables) were analyzed one by one using

**Funding:** University of Gondar, Animal Health Institute (AHI), and International Livestock Research Institute (ILRI) supported for data collection and laboratory investigations. The funders had no role in study design, data collection and analysis, decision to publish, or preparation of the manuscript.

**Competing interests:** The authors have declared that no competing interests exist.

bivariate logistic regression model to identify candidate variables with *P value < 0.25*. Multivariable logistic analysis was used to identify potential associated factors using the candidate variables. A *p value ≤ 0.05* at a 95% confidence interval was statistically significant.

## Result

Among the 428 samples, 7.0% (CI: 4.5–9.3) were confirmed *Campylobacter* species. The prevalence of *C. jejuni* and *C. coli* among under-five children was 5.1% (CI: 3.0–7.0) and 1.9% (CI: 0.7–3.3), respectively. *C. jejuni* (73.3%) was dominant over *C. coli* (26.7%). The resident, contact with domestic animals, and parents/guardians education level were significantly associated with campylobacteriosis among under-five children. One-third of the *Campylobacter* isolates (33.3%, 10/30) were resistant to ciprofloxacin and tetracycline whereas 10.0% (3/30) were resistant to erythromycin. Furthermore, 3.3% (1/30) of the *Campylobacter* were found to be multidrug-resistant.

## Conclusion

The prevalence of *Campylobacter* species was 7.0%. The resistance rate of *Campylobacter* species of ciprofloxacin and tetracycline-resistance strains was 33.3%. Peri-urban residence, contact with domestic animals, and low parental educational statuses were significantly associated factors with increased risk of *Campylobacter* infection. Continuous surveillance on antimicrobial resistance and health education of personal and environmental hygiene should be implemented in the community.

## Introduction

Diarrheal illnesses represent a significant public health concern globally, ranking as the eighth leading cause of death with 1.6 million fatalities. Approximately one-fourth of these deaths occur in children under the age of five. Moreover, over 90% of diarrheal cases are concentrated in South Asia and sub-Saharan nations, where inadequate health infrastructure is prevalent [1]. The global burden of diarrheal illnesses attributed to *Campylobacter* species was estimated to be around 172.33 million, with 88.35 million of these cases affecting children below the age of 5 [2].

*Campylobacter* species are zoonotic bacteria commonly found in food producing animals, wild mammals, and birds. Chickens, in particular, serve as the primary reservoir for *Campylobacter* species and can contribute to the development of drug-resistant strains such as *Campylobacter jejuni* and *C. coli* [3]. Human infections typically occur through the handling or processing of contaminated animal food products, as well as direct contact with infected animals [4]. While chicken production can address income and food insecurity issues, it also poses a potential risk for Campylobacter enteritis, accounting for 50–80% of cases [5].

*Campylobacter* enteritis is a self-limiting disease by which the most vulnerable groups includes individuals younger than five years, people with the age greater than 65 years, and individuals with underline conditions (AIDS and diabetes) [6]. Different study reports showed that the first line of choice for the treatment of Campylobacter gastroenteritis are usually macrolides (erythromycin and azithromycin), whereas floroquinolone (ciprofloxacin) and tetracycline are alternative antibiotics [7]. However, as a result of the increased emergence of resistance to floroquinolone by *Campylobacter* species, macrolides are found as the drug of

choice to treat *Campylobacter* enteritis [8]. Furthermore, there are different study reports that showed the emergence erythromycin resistant *Campylobacter* species elsewhere in the world [9–11].

Furthermore, there have been global reports on the emergence of multi-drug resistance (MDR) *C. jejuni* and *C. coli*. The primary contributing factor to the development of multidrug resistance has also been identified as the irrational use of antibiotics, including self-prescription, as well as their utilization for growth promotion in animal husbandries [12, 13]. Tetracycline resistance occurs when the ribosomal A site is covered by TetO protein. Ciprofloxacin resistance is caused by a point mutation in the GyrA protein within the quinolone resistance-determining region (QRDR). Erythromycin resistance is linked to mutations in the 23S rRNA, as well as the L4 and L22 50S ribosomal proteins loop. Additionally, *Campylobacter* species exhibit multidrug resistance by reducing intracellular concentration through efflux of erythromycin, ciprofloxacin, and tetracycline [14–16].

Detection and identification of *Campylobacter* species is not a common practice in low- and middle-income countries such as Ethiopia due to limited microbiology laboratories, inadequate supplies, and lack of equipment. Therefore, the findings of this research will provide valuable quantitative data on the prevalence of both *Campylobacter jejuni* and *Campylobacter coli*. This data will contribute to a better understanding of the impact of Campylobacter enteritis on children under five years old and will assist in making informed public health decisions. Additionally, knowing the extent of Campylobacter infection aids in allocating appropriate resources for the healthcare system.

In Ethiopia, there have been limited research studies conducted on the antimicrobial resistance pattern among *C. jejuni* and *C. coli* in under-five children with diarrhea. The majority of previous studies in Ethiopia solely relied on conventional culturing techniques. However, the present study utilized an advanced laboratory assay called Matrix-assisted laser desorption ionization–time-of-flight mass spectrometry (MALDI-TOF MS) to enhance the accuracy of *Campylobacter* species identification. Matrix-assisted laser desorption ionization-time of flight mass spectrometry provides a rapid and reliable method to precisely identify a micro-organism from a culture within minutes, unlike the longer timeframes required by conventional methods [17]. Therefore this study aimed to determine the prevalence, associated risk factors, and antimicrobial resistance pattern of *Campylobacter jejuni* and *C. coli* among under-five children with diarrhea.

## Materials and methods

### Study design and study area

A health institution-based prospective cross-sectional study was conducted from 7th November 2022 to 28th April 2023 at Gondar and Bahir Dar City, Northwest Ethiopia. Gondar City is an administrative center of the Central Gondar Zone located 657 km away from Addis Ababa, the capital city of Ethiopia. According to the Gondar City Municipality 2017 report, about 360,600 people live in Gondar City [18]. The City is divided into 24 'kebeles' (the smallest administrative unit), 13 of which are urban and 11 peri-urban. The Gondar City has one government compressive specialized hospital and 7 government health centers. Bahir Dar is the capital city of the Amhara National Regional State which is 488 Km away from Addis Ababa, the capital city of Ethiopia. According to the Bahir Dar health office, the total population of Bahir Dar was estimated around 267,350 by the year 2017. Of these, the under-five children accounted for 11,776, and 7919 of them lived in urban but the other 3857 were in peri-urban parts of the city.

## Source and study population

The source population was all children under-five in Gondar and Bahir Dar presenting with diarrhea at government health centers. The study population was all children under-five with diarrhea attending randomly selected government health centers.

## Sample size determination

The sample size was determined by applying single population formula using previous 15.4% prevalence of *Campylobacter* species among under five diarrheic patients [19] at 95% confidence interval and precision of 5%. The formula applied was: $n = Z^2 p (1-p)/ d^2$. With consideration of design effect, the total sample size was 440.

## Sampling technique

Two sites, Gondar and Bahr Dar, were chosen purposely for the research. Furthermore, Kebeles were listed from urban and peri-urban areas of these two sites and six kebele (3 from urban and 3 from peri-urban) were also randomly selected. A total of under-five aged children (440) were allocated to six randomly selected health facilities. The recruitment of under-five children with diarrhea was done using a convenient sampling technique, which involved selecting those with three or more loose or watery stools in 24 hours.

## Data and sample collection

After securing consent from parents/ guardians, a structured questionnaire was used to collect data related to socio-demographic characteristics, associated risk factors, and clinical information. Data were collected by trained nurses who were working at the respective health institutions. Freshly passed stool samples (5ml) were collected by trained laboratory professionals and the same code was given to the samples and the questionnaire.

## Transportation and processing of the specimen

Once collected, the specimens were placed into universal bottles containing Cary-Blair Transport Medium (Oxoid, Thermo Fischer Scientific, Basingstoke, Hampshire, United Kingdom). The entire specimens were kept in an ice box containing an ice pack with an approximate temperature of 4˚C and were transported to the University of Gondar, College of Medicine and Health Science Microbiology Teaching Laboratory. Each sample was inoculated onto modified Cefoperazone Charcoal Deoxychocolate agar (mCCDA) supplemented with CCDA selective supplement (Oxoid, Thermo Fischer Scientific, Basingstoke, Hampshire, United Kingdom). The inoculated plates were incubated in an anaerobic jar under a microaerophilic environment provided by Gas generating sachets containing 5% $O_2$, 10% $CO_2$, and 85% $N_2$ and incubated at 42˚C for 48 hours.

## Isolation and identification of *Campylobacter* species

After 48 hours of incubation, suspected growth was examined macroscopically for gray, moist spreading, metallic sheen, and discrete colonies. Microscopic examination was performed to observe darting movement and Gram-negative reaction with a curved, 'S' shaped appearance. Catalase and oxidase tests were done for biochemical characterization. Suspected colonies were inoculated and reinoculated on Colombia blood agar (Oxoid, Thermo Fischer Scientific, Basingstoke, Hampshire, United Kingdom) and incubated in a microaerophilic environment at 37 ˚C for 24–48 hours [20]. Then after, the pure colony was harvested and placed in cryo

vials containing Tryptone soya broth supplemented with 20–30% glycerol for shipping to the International Livestock Research Institute (ILRI), Nairobi, Kenya.

## Confirmation by MALDI-TOF MS assay

The bacterial isolates were revived on Colombia blood agar and analyzed by Matrix-assisted laser desorption ionization–time-of-flight mass spectrometry (MALDI-TOF MS) (Bruker Daltonics, Bremen, Germany). Sample preparation was made by smearing a thin film of part of the colony from each isolate on the target plate using a sample applicator. Also, deposit 1µl IVD BTS onto each of the labeled positions and allow to air dry at room temperature. Afterward, 1µl IVD HCCA matrix solution was overlaid on a sample and IVD BTS and was crystallized at room temperature. Finally, the MALDI target plate was loaded into the mass spectrometer for confirmation of the isolates.

## Antimicrobial susceptibility test

Following confirmation of *Campylobacter* species, a modified disc diffusion technique (Kirby-Bauer) was done on BD Mueller Hinton Fastidious Agar (MH-F) according to the European Committee on Antimicrobial Susceptibility Testing (EUCAST) guideline [21]. Around 2–3 pure colonies were added into sterile normal saline and checked with a McFarland densitometer (Grant-bio) to adjust at 0.5 McFarland standard. The sterile applicator cotton swab was deep into the suspension and squeezed against the inside wall of the test tube to avoid heavy inoculation. Then the strain was uniformly plated on the entire BD Mueller Hinton Fastidious Agar (MH-F) (Oxoid, Thermo Fischer Scientific, Basingstoke, Hampshire, United Kingdom). The inoculated plates were kept for 3–5 minutes until air dry. Based on EUCAST recommendation, ciprofloxacin (CIP, 5 µg), erythromycin (ERY, 15 µg) and, tetracycline (TET, 30 µg) discs were placed on the MH-F plate using a disc dispenser. All the plates then were placed into an anaerobic jar containing gas-generating sachets and incubated at 42˚C for 24hours. After 24hours incubation, the diameter of the zone of inhibition was measured by a ruler. Multidrug-resistant strains were determined by using previously set criteria [22].

## Data analysis

The outcome variable was Campylobacter infection among children aged under-five years. All data were directly entered and analyzed using SPSS version 25. Descriptive analysis was made to compute frequency and proportions. The outcome variable was Campylobacter infection among children aged under-five years. Odds ratio was calculated to see the association between dependent and independent variables. All potential covariates (independent variables) were analyzed one by one using bivariate logistic regression model to identify candidate variables with *P value < 0.25*. Multivariable logistic analysis was used to identify potential associated factors using the candidate variables. The model fitness was checked using the Hosmer-Lemeshow goodness-of-fit test. A *p value ≤ 0.05* at a 95% confidence interval was statistically significant.

## Data quality assurance

Trained nurses and Laboratory professionals were recruited to interview the parents/guardians and sample collection from children, respectively. The pretest was carried out on 5% of the total questionnaire before any actual data collection. The standard operational procedure is followed during any laboratory analysis. The sterility test was made on 5% of prepared media by incubating for 1–2 days depending on the media type. Furthermore, antimicrobial disc

quality was checked using control strains (*Campylobacter jejuni* NCTC 11351 and *Campylobacter coli* NCTC 11351). Bacterial test standard (BTS) was used to check the proper functionality of the MALDI-TOF-MS assay. The overall laboratory quality was maintained by adhering to pre-analytic, analytical, and post-analytical quality control.

### Ethical considerations

Ethical clearance was obtained from the University of Gondar Institutional Review Board (IRB) with ethical consideration number (V/P/RCS/05/2715/2021). A written informed consent was obtained from participants' parents/guardians after explaining the purpose and objective of the study. Participants had a full right to continue or withdraw from the study. All information was kept confidential by assigning code and assessed by the principal investigator and supervisors. The laboratory results were communicated to concerning stakeholders and participants. When the stool sample was positive for Campylobacter species, we communicated with concerned health professionals and treated the patients according to WHO management guidelines.

## Results

### Socio-demographic characteristics of under-five children

Out of the 440 sample size initially proposed, a total of 428 study participants were included in this study, resulting in a respondent rate of 97.3%. The remaining twelve individuals may have either declined to participate or were unable to provide the required specimen. Among the under-five children, males were 219 (51.2%) and females were 209 (48.8%). The majority of the under-five children were urban dwellers. More than three-fourths (78.5%) of the parents had formal education. Nearly one-third of the under-five children had contact with domestic animals. Data on hand washing habits showed that 20.8% of the parents had the practice of hand washing only sometimes after toilets and 19.6% before feeding. In this study, 16.4% of the parents used only water for cleaning utensils [Table 1].

### Clinical characteristics of children aged under-five years

Among the 428 under-five children with diarrhea, 204 (52.3%) had fever, 245 (57.2%) had vomiting, and 245 (57.2%) had abdominal pain. The majority (92.3%) of the under-five children experienced diarrhea that lasted up to five days and 61.9% of them had mucoid and bloody diarrhea [Table 2].

### Environmental characterization of under-five children

According to the findings of the current study, it is evident that a significant proportion of children under the age of five (70.8%) reside in houses with cemented floors. Additionally, a majority of these houses (89.7%) are equipped with latrines. However, it is concerning to note that approximately one-fourth (22.9%) of the parents or guardians have inadequate waste disposal systems. On a positive note, the majority of the participants in the study obtain their drinking water from protected sources. Conversely, a large majority (81.1%) do not practice any form of home-based water treatment. Furthermore, the data reveals that 84.6% of the study participants have a water source that requires a maximum of 30 minutes to access [Table 3].

**Table 1. Socio-demographic characteristics of the under five children and their parents/guardians in the Amhara National Region state, Northwestern Ethiopia, November, 2022 to April, 2023.**

| Variables | Category | Frequency (%) | *Campylobacter* infection (%) |
|---|---|---|---|
| Sex | Male | 219 (51.2) | 16 (7.3) |
| | Female | 209 (48.8) | 14 (6.7) |
| Age in month | 0–6 | 52 (12.1) | 1 (1.9) |
| | 7–59 | 376 (87.9) | 29 (7.7) |
| Residence | Urban | 278 (65.4) | 7 (2.5) |
| | Peri-Urban | 150 (34.6) | 23 (15.3) |
| Contact with domestic animals | Yes | 98 (22.9) | 18 (18.4) |
| | No | 330 (77.1) | 12 (3.6) |
| Parent relation to a child | Mother | 370 (86.4) | 27 (7.3) |
| | Father | 53 (12.4) | 2 (3.8) |
| | Guardian | 5 (1.2) | 1 (20.0) |
| Education status of parent | No formal Education | 129 (30.1) | 15 (11.6) |
| | Formal Education | 299 (69.9) | 15 (5.0) |
| Number of children | Only one | 149 (34.8) | 19 (7.4) |
| | > one | 279 (65.2) | 11 (6.8) |
| Hand washing after Latrine | Yes, always | 335 (78.3) | 26 (7.8) |
| | Yes, some time | 89 (20.8) | 3 (3.4) |
| | Not at all | 4 (0.9) | 1 (25.0) |
| Hand washing habit before feeding and preparation | Yes, always | 340 (79.4) | 25 (7.4) |
| | Yes, some time | 83(19.4) | 4 (4.8) |
| | Not at all | 4 (0.9) | 1 (25.0) |
| Cleaning utensil with | Water and soap | 358 (83.6) | 22(6.1 |
| | Water only | 70 (16.4) | 8 (11.4) |

CI, confidence interval

## Prevalence of *Campylobacter* species among under five children

Among the thirty-two culture positive preliminary isolates, 30 were confirmed as *Campylobacter* species using MALDI-TOF mass spectrometry assay. The prevalence of *Campylobacter* species among under-five children with diarrhea was found to be 7.0% (CI: 4.5–9.3). Furthermore

**Table 2. Clinical characteristics of the under five children with diarrhea in the Amhara National Region state, Northwest Ethiopia, November, 2022 to April, 2023.**

| Variables | Category | Frequency (%) | Campylobacter infection (%, CI) |
|---|---|---|---|
| Fever | Yes | 204(52.3) | 10 (4.9) |
| | No | 224 (47.7) | 20 (8.9) |
| Vomiting | Yes | 245 (57.2) | 19 (7.8) |
| | No | 183 (42.8) | 11 (6.0) |
| Abdominal pain | Yes | 332 (77.6) | 24 (7.2) |
| | No | 96 (22.4) | 6 (6.3) |
| Duration diarrhea | 1-5days | 395 (92.3) | 29 (7.3) |
| | >5days | 33 (7.6) | 1 (3.0) |
| Consistency | Watery | 163 (38.1) | 8 (4.9) |
| | Mucoid & bloody | 265 (61.9) | 22 (8.3) |

CI, confidence interval

**Table 3. The environmental condition of the under five children with diarrhea in the Amhara region, Northwest Ethiopia, November, 2022 to April, 2023.**

| Variables | Category | Frequency (%) | Campylobacter infection (%, CI) |
|---|---|---|---|
| Floor of the house | Mud | 125 (29.2) | 5 (4.0) |
| | Cemented | 303 (70.8) | 25 (8.3) |
| Availability of latrine | Yes | 384 (89.7) | 24 (6.3) |
| | No | 44 (10.3) | 6 (13.6) |
| Waste disposal system | Proper | 330 (77.1) | 21(6.4) |
| | Open field | 98 (22.9) | 9 (9.2) |
| Source of drinking water | Protected | 362 (84.6) | 26 (7.2) |
| | Unprotected | 66 (15.4) | 4 (6.1) |
| Home based water treatment | Yes | 81 (18.9) | 2 (2.5) |
| | No | 347 (81.1) | 28 (8.1) |
| Distance of water source | Up to30 minutes | 363 (84.6) | 19 (5.2) |
| | >30 minutes | 65 (15.4) | 11(16.9) |

CI, confidence interval

the prevalence of *C. jejuni* was 5.1% (CI: 3.0–7.0) and the prevalence of *C. coli* was 1.9% (CI: 0.7–3.3). *C. jejuni* was more commonly found among the *Campylobacter* species, accounting for 73.3% of cases, while *C. coli* accounted for 26.7%. The prevalence of *Campylobacter* species was similar among male and female participants (*p value = 0.81*). Although the prevalence of *Campylobacter* species was higher among children aged 7–59 months, this difference was not statistically significant. However, the prevalence of *Campylobacter* species was higher among children from peri-urban areas (*p value = 0.001*) and those who had contact with domestic animals (*p value = 0.015*) [Table 4].

## Potential associated factors and their relationship with *Campylobacter* enteritis

The results of the present study indicated that 8 variables examined through Bivariate logistic regression had a *P value* < 0.25, and thus were selected for further investigation using a multivariable logistic regression model. Out of the eight variables, residence, contact with domestic animals, and the educational status of the parents/guardians were found to be statistically significant (p<0.05). Consequently, children residing in peri-urban areas and having contact with domestic animals were 4.91 (95% CI: 1.92–12.54, *p value = 0.001*) and 5.03 (95% CI: 2.18–11.62, *p value = 0.001*) times more likely to contract Campylobacter gastroenteritis. Moreover, children under the age of five whose parents/guardians had no formal education

**Table 4. The prevalence of *Campylobacter* species among under five children with diarrhea in Amhara National Regional state, Northwest Ethiopia, November, 2022 to April, 2023.**

| Strain | Campylobacter infection | |
|---|---|---|
| | Positive (%: CI) | Negative (%) |
| *Campylobacter species* | 30 (7.0%: 4.5–9.3) | 398 (93.0) |
| • *Campylobacter jejun* | 22(5.1%: 3.0–7.0) | 406 (94.9) |
| • *Campylobacter coli* | 8 (1.9%: 0.7–3.3) | 420 (98.1) |

CI, confidence interval

**Table 5. Bivariate and multivariable logistic regression analysis of factors associated with *Campylobacter jejuni* and *Campylobacter coli* infection among under-five children with diarrhea at the Amhara National Regional state, Northwest Ethiopia, November, 2022 to April, 2023.**

| Predictors | Campylobacter infection | | COR (95% CI) | *P value* | AOR (95% CI) | *P value* |
|---|---|---|---|---|---|---|
| | Negative (%) | Positive(%) | | | | |
| **Age in month** | | | | | | |
| 0–6 | 51 | 1 | 1 | | 1 | |
| 7–59 | 348 | 29 | 5.158 (0.546–30.81) | 0.158 | 5.16 (0. .624–42.65) | 0.164 |
| **Residence** | | | | | | |
| Urban | 272 | 8 | 1 | | 1 | |
| Peri-urban | 126 | 22 | 7.01 (2.93–16.77) | 0.001 | 4.91 (1.92–12.54)* | **0.001** |
| **House flour type** | | | | | | |
| Mud | 120 | 5 | 0.48(0.18–1.30) | 0.125 | 0. .62 (0. .20–1.89) | 0.441 |
| Cemented | 279 | 25 | 1 | | 1 | |
| **Availability of latrine** | | | | | | |
| Yes | 38 | 6 | 1 | | 1 | |
| No | 360 | 24 | 2.48 (0.95–6.46) | 0.07 | 1.36 (0.45–4.14) | 0.564 |
| **Contact with domestic animals** | | | | | | |
| Yes | 80 | 18 | 5.96 (2.76–12.88) | 0.001 | 5.03 (2.18–11.62)* | **0.001** |
| No | 318 | 12 | 1 | | 1 | |
| **Home based water treatment** | | | | | | |
| Yes | 79 | 2 | 1 | | 1 | |
| No | 309 | 28 | 3.33 (0.78–14.31) | 0.094 | 2.53 (0.53–12.19) | 0.274 |
| **Education status of parent** | | | | | | |
| No formal education | 110 | 20 | 3.51 (1.65–7.42) | 0.001 | 3.14 (1.36–7.25)* | **0.008** |
| Formal education | 288 | 10 | 1 | | 1 | |
| **Cleaning utensils with** | | | | | | |
| Water only | 62 | 8 | 2.10 (0.88–4.88) | 0.119 | 1.47 (0.547–3.97) | 0.417 |
| Water and soap | 337 | 22 | 1 | | 1 | |

**COD**- crude odds ratio, **AOR**- adjusted odds ratio, **CI**- confidence interval,

* statistically significant at p < 0.05

were 3.14 (95% CI: 1.36–7.25, *p value = 0.008*) times more likely to be infected with *Campylobacter* species [Table 5].

## Antimicrobial resistance patterns

A total of 30 confirmed *Campylobacter* species were tested to determine their susceptibility to antimicrobial agents. The antimicrobial agents used in the study were ciprofloxacin, erythromycin, and tetracycline, as recommended by the European Committee for antimicrobial susceptibility testing (EUCAST) guideline. It was found that one-third of the *Campylobacter* isolates (33.3%, 10/30) were resistant to both ciprofloxacin and tetracycline. Additionally, 10.0% (3/30) of the *Campylobacter* species showed resistance to erythromycin. The resistance levels of *C. jejuni* and *C. coli* to ciprofloxacin, erythromycin, and tetracycline were as follows: (22.70%,5/22 and 62.50%, 5/8), (4.50%, 1/22 and 12.50%, 1/8), and (27.30%, 6/22 and 48.00%, 3/8), respectively. None of the *Campylobacter* species demonstrated susceptibility to all three antibiotics. Notably, the resistance rate to ciprofloxacin and tetracycline was higher in *C. coli* (62.5% and 48.0%, respectively) compared to *C. jejuni* (22.7% and 27.3%, respectively) (p <0.05) [Fig 1].

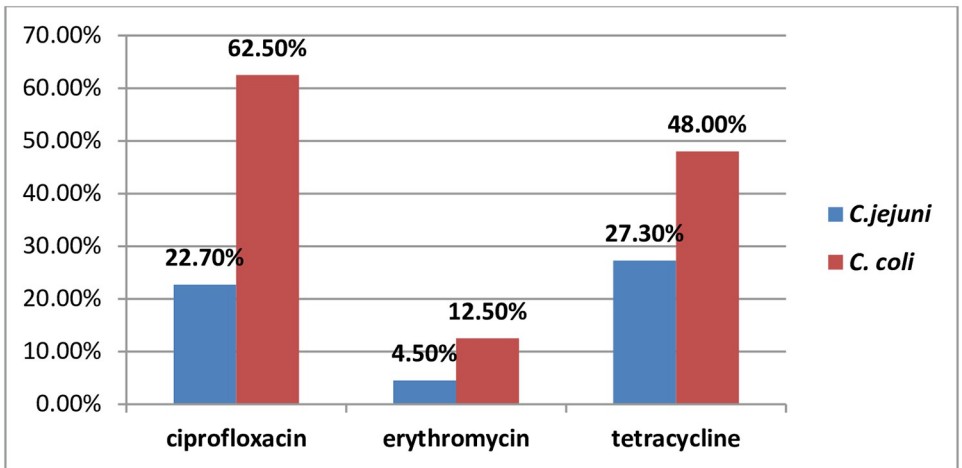

**Fig 1. Antimicrobial resistance pattern of *C. jejuni* and *C. coli* among under-five children with diarrhea in Amhara Region, Northwest Ethiopia, November, 2022 to April, 2023.**

## Multidrug resistance

In this study, nearly one-third (33.3%) of the *Campylobacter* species isolates showed resistance to more than two antimicrobial agents. Three *Campylobacter* isolates were resistant to ciprofloxacin and erythromycin, five isolates were resistant to ciprofloxacin and tetracycline, but only one isolate was resistant to erythromycin and tetracycline. On the other hand, only 1 *Campylobacter* species isolate was resistant to ciprofloxacin, erythromycin, and tetracycline (multidrug-resistant isolate) [Table 6].

## Discussion

The objective of this research was to identify Campylobacter gastroenteritis, related factors, and the antimicrobial resistance patterns of *Campylobacter jejuni* and *C. coli* in children under the age of five with diarrhea. *Campylobacter jejuni* was identified as the primary cause of Campylobacter gastroenteritis in the study. Factors such as place of residence, exposure to domestic animals, and the educational background of parents/guardians were found to be associated with these infections. The lowest level of resistance among *Campylobacter* species was observed in erythromycin, while *Campylobacter coli* exhibited higher resistance to ciprofloxacin and tetracycline.

**Table 6. Distribution of multidrug resistant profile of *Campylobacter species* among under five children with diarrhea in Amhara National Regional state, Northwest Ethiopia, November, 2022 to April, 2023.**

| Resistance profile | *Campylobacter* spp (n = 30) | *C. jejuji* (n = 22) | *C. coli* (n = 8) |
|---|---|---|---|
| CIP^R^-ERY^R | 3 (10.0%) | 1(4.5%) | 2 (25.0%) |
| CIP^R^-TET^R | 5 (16.7%) | 2(9.1%) | 3 (37.5%) |
| ERY^R^-TET^R | 1 (3.3%) | 0 | 1(12.5%) |
| ERY^R^-CIP^R^-TET^R | 1 (3.3%) | 0 | 1(12.5%) |
| **Total** | **10 (33.3%)** | **3 (13.6%)** | **7 (87.5)** |

CIP^R^-ERY^R, ciprofloxacin-erythromycin resistant, CIP^R^-TET^R, ciprofloxacin-tetracycline resistant, ERY^R^-TET^R, erythromycin-tetracycline resistant, ERY^R^-CIP^R^-TET^R, erythromycin-ciprofloxacin-tetracycline resistant

The prevalence of *Campylobacter* species among under-five children with diarrhea in this study was found to be 7.0%, which aligns with similar reports from Hawassa, Ethiopia, 6.8% [23], and Brazil, 6.87% [24]. However, our findings showed higher rates compared to studies in Nairobi, Kenya, 4.8% [25], and another study from Nairobi city, 1.6% [26]. Conversely, the prevalence in our study was lower than reports from Southwest Ethiopia, 8.9% [27], eastern Ethiopia, 8.43% [28], Khartoum, Sudan, 11.0% [29], Morogoro, Tanzania, 19.0% [30], Dhaka, Bangladesh, 23.7% [31], central Iran, 33.0% [32], Northwest Province, South Africa, 33.0% [33], Guinea-Bissau, 53.1% [34], and Latvia, 72.0% [35]. The differences in findings across studies could be attributed to factors such as personal and environmental hygiene, contact with domestic animals, study duration, seasonal and geographical variations, sample size, and laboratory methods.

Our study also revealed that the prevalence of *C. jejuni* among under-five children was 5.1%. The current result was comparable with a report from Sudan, 4.0% [36]. However, our finding was higher than the study conducted in Indonesia, 3.08% [37]. On the other hand, our finding was lower than reports from the Southern Equator, 7.9% [38], Kenya, 12.9% [14], Bulgaria, 33.2% [39] and Pakistan, 48.2%, [40]. The probable reasons for the difference between the current finding and previous reports might be due to the degree of contact with domestic animals, poor food handling, consumption of undercooked meat, poor personal and environmental hygiene, and the laboratory method used.

In this study, C. *jejuni* was the most dominant pathogen (n = 22; 73.3%) than *C. coli (n = 8;* 26.7). Previous reports from different African countries also showed the dominance of *C.jejuni* over *C. coli.* In Kenya an 80.8% *C. jejuni* and 26.9% *C. coli* [25], In South Africa 45.0% *C. jejuni* and 37.0% *C. coli* [33], and in Egypt 61.0% *C. jejuni* and 37.0% *C.coli* [41] were reported. One contributing factor for the higher frequency of *C. jejuni* than *C.coli* might be the ability of *C. jejuni* to survive in the environment for a longer time than *C. coli.* Nilsson *et al* reported that *C. jejuni* better survived than *C. coli* in the water environment [42]. Bronowski *et al* reported that *C. jejuni* can survive through mechanisms such as exhibiting aerotolerance, resistance to starvation, biofilm formation, and converting to non-culturable viable forms [43].

The recent research revealed a significant link between the living arrangements of children under the age of five and Campylobacter infection. This could be attributed to the backyard production system in peri-urban areas, which may contribute to contamination on the shared ground where children play. Additionally, it was found that in peri-urban communities where people live in the same house as animals, there is a higher likelihood of infection in children [25]. Our study also found that contact with domestic animals was a contributing factor to Campylobacter gastroenteritis, which is supported by similar studies conducted in Jimma, Ethiopia [27] and Hawassa, Ethiopia [23]. Furthermore, the educational level of the parents or guardians was found to be significantly associated with Campylobacter gastroenteritis in children under the age of five. Specifically, children whose mothers had no formal education were more likely to have Campylobacter gastroenteritis. Other studies have also reported a significant association between parent educational level and diarrheal disease [4, 44].

Our research uncovered that 33.3% of *Campylobacter spp* exhibited resistance to ciprofloxacin, a percentage higher than that reported in Nairobi, Kenya (0.0%), South Africa (18%), and Hawassa, Ethiopia (18.7%) [23, 25, 33]. Conversely, our findings showed a lower resistance rate compared to the study conducted in central Iran, which reported a resistance rate of 71.1% [32]. Furthermore, 10.0% of the *Campylobacter* species were resistant to erythromycin, contrasting with the report from Hawassa, Ethiopia, where none of the species exhibited resistance to erythromycin [24]. However, higher rates of erythromycin-resistant *Campylobacter* were observed in South Africa (18.0%), Nairobi, Kenya (50.0%), and Central Iran (68.9%)

[26, 32, 33]. Additionally, our study found that 33.3% of *Campylobacter* species were resistant to tetracycline, consistent with the findings of the study conducted in South Africa by Shobo et al., which reported a resistance rate of 33.3% [45].

In the present investigation, it was observed that 62.5% of *C. coli* exhibited resistance to ciprofloxacin, while *C. jejuni* showed a resistance rate of 22.70%. Our findings align with a study conducted by Ford et al in the United States of America, which reported a higher prevalence of ciprofloxacin-resistant *C. coli* (39.6%) compared to *C. jejuni* (28.4%) [46]. Additionally, our study revealed that the resistance level of *C. coli* (12.5%) to erythromycin was higher than that of *C. jejuni* (4.5%). This finding is consistent with a study conducted in Taiwan, which also highlighted the dominance of ciprofloxacin-resistant C. coli (62.2%) over *C. jejuni* (2.2%) [47]. Furthermore, our research demonstrated that the rate of tetracycline resistance in *C. coli* (48.0%) was higher than that in *C. jejuni* (27.3%). A similar study also reported a higher resistance level of *C. coli* (55.6%) to tetracycline compared to *C. jejuni* (25.9%) [47]. The higher resistance rate of *C. coli* compared to *C. jejuni* in these three antibiotics may be attributed to its increased potential for acquiring resistance genes through horizontal gene transfer [48, 49].

Multidrug resistant *Campylobacter* species is a strain resistant to macrolides (erythromycin), chloeoquinolone (ciprofloxacin) and Tetracycline (tetracycline). The emergence of multidrug resistant *Campylobacter* species poses human health risk as limiting treatment options. In the present study 3.3% of the strains were multidrug resistant *Campylobacter* species. This might be predicted that *cmeABC* gene that code efflux complex on cell membrane which promoting resistance to structurally different antibiotics such as erythromycin, ciprofloxacin and tetracycline [50–52].

The data obtained from this research must be analyzed with the following limitations in mind. Initially, the study was carried out in specific regions of Ethiopia and may not be indicative of the entire nation. Additionally, the duration of the study was six months, potentially impacting the prevalence of *Campylobacter* species.

## Conclusion

The prevalence of *Campylobacter* species among under-five children was 7.0%. Data also showed high rate of tetracycline and ciprofloxacin resistance among *Campylobacter* species. The frequency of antimicrobial resistance was higher in *Campylobacter coli* compared with *Campylobacter jejuni*. *Campylobacter* species caused gastroenteritis in general and drug resistant strains in particular could be potential treat among under-five children suffering from diarrheal disease. Therefore, culture and sensitivity test should be done on routine bases prior administration of medications. Living together with domestic animals could be a potential source for *Campylobacter* species and therefore awareness through health education can reduce the risk of acquiring the pathogen.

## Acknowledgments

The authors would like to acknowledge the University of Gondar for all the support given to us. Our gratitude also goes to the International Livestock Research Institute (ILRI) specifically the One Health Research in Africa (OHRECA) for the support given to conduct advanced laboratory works in Nairobi, Kenya. We also thank the Ethiopia Animal Health Institute (AHI) for the support given to run laboratory works and storage of samples. We thank all study participants for being willing to provide data and samples for this study.

## Author Contributions

**Conceptualization:** Mesfin Worku, Belay Tessema, Getachew Ferede, Delia Grace, Baye Gelaw.

**Data curation:** Mesfin Worku, Belay Tessema, Getachew Ferede, Arshnee Moodley, Delia Grace, Baye Gelaw.

**Formal analysis:** Mesfin Worku, Belay Tessema, Getachew Ferede, Arshnee Moodley, Delia Grace, Baye Gelaw.

**Funding acquisition:** Mesfin Worku.

**Investigation:** Mesfin Worku, Getachew Ferede, Linnet Ochieng, Shubisa Abera Leliso, Arshnee Moodley, Delia Grace, Baye Gelaw.

**Methodology:** Mesfin Worku, Belay Tessema, Getachew Ferede, Arshnee Moodley, Delia Grace, Baye Gelaw.

**Project administration:** Mesfin Worku.

**Resources:** Florence Mutua, Arshnee Moodley, Delia Grace, Baye Gelaw.

**Supervision:** Arshnee Moodley, Delia Grace, Baye Gelaw.

**Validation:** Delia Grace, Baye Gelaw.

**Writing – original draft:** Mesfin Worku.

**Writing – review & editing:** Getachew Ferede, Florence Mutua, Arshnee Moodley, Delia Grace, Baye Gelaw.

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
