## [Decision Letter · Decision Letter 0]

12 Mar 2024

PONE-D-24-04047Campylobacter jejuni and Campylobacter coli, determinants and antimicrobial resistance pattern among under-five children with diarrhea in Amhara National Regional state, Northwest Ethiopia.PLOS ONE

Dear Dr. Worku,

Thank you for submitting your manuscript to PLOS ONE. After careful consideration, we feel that it has merit but does not fully meet PLOS ONE’s publication criteria as it currently stands. Therefore, we invite you to submit a revised version of the manuscript that addresses the points raised during the review process.

We look forward to receiving your revised manuscript.

Kind regards,

Eric Ogola, BVM,MPH,CASMI Fellow

Academic Editor

PLOS ONE

Journal Requirements:

   "University of Gondar, Animal Health Institute (AHI), and International Livestock Research Institute (ILRI) supports for data collection and laboratory investigations."

4. We note that your Data Availability Statement is currently as follows: All relevant data are within the manuscript and its Supporting Information files

7. Please ensure that you refer to Figure 1 in your text as, if accepted, production will need this reference to link the reader to the figure.

8. Please include a caption for figure 1. 

9. We note you have included a table to which you do not refer in the text of your manuscript. Please ensure that you refer to Table 6 in your text; if accepted, production will need this reference to link the reader to the Table.

Reviewers' comments:

Reviewer's Responses to Questions

**Comments to the Author**

1. Is the manuscript technically sound, and do the data support the conclusions?

Reviewer #1: Yes

Reviewer #2: Yes

2. Has the statistical analysis been performed appropriately and rigorously? 

Reviewer #1: Yes

Reviewer #2: Yes

3. Have the authors made all data underlying the findings in their manuscript fully available?

Reviewer #1: Yes

Reviewer #2: No

4. Is the manuscript presented in an intelligible fashion and written in standard English?

Reviewer #1: Yes

Reviewer #2: No

5. Review Comments to the Author

Reviewer #1: Line 45: Among the Campylobacter isolates, C. jejuni were commonly isolate (73.3%).......

Kindly add 95%CI to all proportions in the manuscript either in the text or the tables. This provide a better expression of the proportion uncertainty.

Reviewer #2: This is a very important manuscript presenting the results of a study that assessed the prevalence of Campylobacter spp among children aged under-five years with diarrhoea. This study has many strengths including:

1- It is addressing a zoonotic disease that is opportunistic in nature

2- The study used advanced assay techniques (MALDI-TOF MS) to identify the organisms and included antibiotic susceptibility testing.

3- The sample size is fairly large

However, the manuscript has some important weaknesses that should be addressed before publication as summarised below:

- Language - there are numerous grammatical errors in the manuscript that are distracting from the important message. The authors could seek professional language editing services to improve the quality of this manuscript.

- The abstract methods do not include some crucial information about how the study was conducted and only focusses on laboratory methods.

Introduction - although the authors mention that they used an advanced laboratory technique (MALDI-TOF MS) to identify the organisms, they do not mention what is the advantage of this method over traditional methods.

Methods - source population - the two sentences in this section are very similar and repetitive.

Sampling technique - it is not very clear how the children were selected and what case definitions of diarrhoea were applied.

Data analysis - the cut off for including variable in a multivariate model is given as 0.25. This threshold is very high, the authors should consider repeating this analysis using a lower threshold.

Results - There are two Table 4s in this section. One appears after Table 6. The abbreviations used in the second Table 4 should be spelt out in a footnote below the table.

Discussion - the authors should revise the first paragraph in this section to summarise the main findings in the study.

The authors note that the prevalence of campylobacter spp was similar to that found in other studies in Ethiopia but higher than other countries such as Kenya. Could this be attributed to the different laboratory techniques? In general, this section focusses so much on comparing study findings with previous study at the expense of explaining the implications of their findings for public health.

conclusions - the following statement is not fully backed up by data presented in this manuscript:

"Culture and sensitivity test should be carried out prior to medication of diarrheal patients"

6. PLOS authors have the option to publish the peer review history of their article (what does this mean?). If published, this will include your full peer review and any attached files.

Reviewer #1: No

Reviewer #2: No

---

## [Author Response · Author response to Decision Letter 0]

2 Apr 2024

We appreciate the assessment carried out by editorial team. We reviewed and made necessary correction as follows 

Response: Thank you! we reviewed it again and amended as per PLOS ONE's style requirements

Response: Thank you! We removed funding related text from the manuscript

 "University of Gondar, Animal Health Institute (AHI), and International Livestock Research Institute (ILRI) supports for data collection and laboratory investigations."

Please state what role the funders took in the study. 

Response: Thank you! We amended the financial disclosure as “University of Gondar, Animal Health Institute (AHI), and International Livestock Research Institute (ILRI) supports for data collection and laboratory investigations. The funders had no role in study design, data collection and analysis, decision to publish, or preparation of the manuscript”. 

4. We note that your Data Availability Statement is currently as follows: All relevant data are within the manuscript and its Supporting Information files

Response: Thank you! Sorry for encountered mistake. We have no supporting files and we stated it as “All relevant data are within the manuscript” 

Response: Thank you! We removed ethics statement from other section and stated only in method section. 

7. Please ensure that you refer to Figure 1 in your text as, if accepted, production will need this reference to link the reader to the figure. 

Response: Thank you! We have already stated in Line No 273

8. Please include a caption for figure 1. 

Response: Thank you! We have already stated in Line No 277. But changed the word ‘picture’ to ‘figure’

9. We note you have included a table to which you do not refer in the text of your manuscript. Please ensure that you refer to Table 6 in your text; if accepted, production will need this reference to link the reader to the Table.

Response: Thank you! We corrected the order of each table and referred accordingly 

Response: Thank you! We carefully reviewed and all listed references are complete and cited. We had no retracted papers

Reviewer #1

We would like to express our gratitude to reviewer 1 for providing the important comments on the issue of confidence interval

1. Line 45: Among the Campylobacter isolates, C. jejuni were commonly isolate (73.3%).......

Kindly add 95%CI to all proportions in the manuscript either in the text or the tables. This provides a better expression of the proportion uncertainty.

Response: Thank you so much and we appreciate the concern! We calculated Confidence interval (CI) and added in both text and tables which indicated with yellow color highlight, but 73.3% and 26.7% were calculated to show the frequency of specious among 30 isolates. 

Reviewer #2

We express our gratitude to reviewer 2 for providing a thorough overview of our research and highlighting both its merits and drawbacks. We are committed to improving our manuscript by implementing the required revisions. We have taken the time to provide a detailed response that addresses each raised concern.

1. Language - there are numerous grammatical errors in the manuscript that are distracting from the important message. The authors could seek professional language editing services to improve the quality of this manuscript.

Response: Thank you! We have identified the problem and made corrections at both the word and paragraph levels. Furthermore, we have taken punctuation into account. The highlighted portions are in yellow for your review.

2. The abstract methods do not include some crucial information about how the study was conducted and only focusses on laboratory methods.

Response: Thank you! We included the missed information methods of the abstract and highlighted with yellow color. The number of the word for abstract limited us. (37-39)

3. Introduction - although the authors mention that they used an advanced laboratory technique (MALDI-TOF MS) to identify the organisms, they do not mention what is the advantage of this method over traditional methods.

Response: Thank you! We included about advantage of the MALDI-TOF (Line no 104-106)

4. Methods - source population - the two sentences in this section are very similar and repetitive.

Response: Thank you and accepted and we now clearly stated to avoid similarity (Line 128-130)

5. Sampling technique - it is not very clear how the children were selected and what case definitions of diarrhoea were applied.

Response: Thank you! We restated and kindly refer to Line No 137-142

Data analysis - the cut off for including variable in a multivariate model is given as 0.25. This threshold is very high; the authors should consider repeating this analysis using a lower threshold.

Response: Thank you and we accepted and reviewed. After conducting a bivariate logistic regression analysis, we recalculated our candidate variable and were pleased to find that the results remained unchanged, with all variables falling below 0.2. This led us to replace 0.25 with 0.2.

Results - There are two Table 4s in this section. One appears after Table 6. The abbreviations used in the second Table 4 should be spelt out in a footnote below the table.

Response: Thank you and accepted and corrected accordingly 

6. Discussion - the authors should revise the first paragraph in this section to summarise the main findings in the study.

Response: Thank you! We revised the first paragraph accordingly, Line No 326-333

The authors note that the prevalence of campylobacter spp was similar to that found in other studies in Ethiopia but higher than other countries such as Kenya. Could this be attributed to the different laboratory techniques? In general, this section focusses so much on comparing study findings with previous study at the expense of explaining the implications of their findings for public health.

Response: Thank you and we appreciate your constructive comment. There are various potential factors that may contribute to the disparity between our findings and previous studies. Among these factors, the laboratory method could be one of the probable reasons due to its specificity and sensitivity. The utilization of state-of-the-art technology can significantly impact the prevalence of campylobacters. Additionally, we have also mentioned other potential factors that could account for the variation the prevalence of Campylobacter infections. (L341-344)

Conclusions - the following statement is not fully back up by data presented in this manuscript:

"Culture and sensitivity test should be carried out prior to medication of diarrheal patients"

Response: Thank you and we accept the commend and corrected accordingly (L416-424)

---

## [Decision Letter · Decision Letter 1]

23 Apr 2024

PONE-D-24-04047R1Campylobacter jejuni and Campylobacter coli infection, determinants and antimicrobial resistance pattern among under-five children with diarrhea in Amhara National Regional state, Northwest Ethiopia.PLOS ONE

Dear Dr. Worku,

Thank you for submitting your manuscript to PLOS ONE. After careful consideration, we feel that it has merit but does not fully meet PLOS ONE’s publication criteria as it currently stands. Therefore, we invite you to submit a revised version of the manuscript that addresses the points raised during the review process.

We look forward to receiving your revised manuscript.

Kind regards,

Tebelay Dilnessa, MSc

Academic Editor

PLOS ONE

Journal Requirements:

**Additional Editor Comments:**
The title,* ‘Campylobacter jejuni* and *Campylobacter coli* infection, determinants and antimicrobial resistance pattern among under-five children with diarrhea in Amhara National Regional state, Northwest Ethiopia’. The word ‘resistance’ better be replaced by ‘susceptibility’. Then modify the objective and description of tables accordingly.The affiliation better which was ‘^5^Animal Health Institute, Sebeta, Ethiopia’ should be linked to the author (s) or removed.The email addresses of authors were not needed here in the main manuscript as it appears in the system except the corresponding author.Lines 39 &40: ‘Suspected colonies were analyzed using matrix-assisted laser desorption ionization-time of flight mass spectrometry to confirm the species.’ It is better written as, ‘The suspected colonies were analyzed using matrix-assisted laser desorption ionization-time of flight mass spectrometry (MALDI-TOF-MS) to confirm the species.’Lines: 41&42: ‘Logistic regression was used to analyze the associated factors.’  It was better written as, ‘Logistic regression was used to analyze the associated factors with P-value <0.05’.Lines 43-45: ‘Among the 428 samples, 30 (7.0%) (CI: 4.5-9.3%) were culture-positive for *Campylobacter* species. The prevalence of *C. jejuni* and *C. coli* among under-five children was 5.1% (CI: 3.0-44 7.0%) and 1.9% (CI: 0.7-3.3%)’. Here the sign (%) from the CI part should be removed because by default it has already known. Similarly, the number of isolates with the total better written together with the percentages. This works for the whole abstract and result part of the paper. For example, for the above better written as, “Among the 428 samples, 30 (7.0%) (CI: 4.5-9.3) were culture-positive for *Campylobacter* species. The prevalence of *C. jejuni* and *C. coli* among under-five children was 22/428 (5.1%) (CI: 3.0-44 7.0) and 8/428 (1.9%) (CI: 0.7-3.3)’.Line 64: ‘………health infrastructure is prevalent. [1].’ Better written as, ‘………health infrastructure is prevalent [1].’Line 88: ‘……… in animal husbandries. [12,13].’ Better written as, ‘,… in animal husbandries [12,13].’Similarly, line 94: ‘……………..and tetracycline. [14–16].’  Better written as, ‘…….. and tetracycline [14–16].Line 96 &97: ‘Binary and multivariable logistic analysis was used to identify associated risk factors. A P value less than 0.05 at a 95% confidence interval was statistically significant’. Better written as ‘Bivariable and multivariable logistic regression analysis was used to identify associated risk factors. A P-value < 0.05 at a 95% confidence interval was considered as statistically significant’.In the data analysis part, it is better you add something about the interpretation of ‘MALDI-TOF MS assay.  Line 100: ‘MALDI-TOF MS provides a rapid…….’ It is better written as, ‘Matrix-assisted laser desorption ionization-time of flight mass spectrometry provides a rapid…….’ As the start of a sentence will not be acronyms/abbrevations.Line 100: ‘…….Campylobacter species identification.’ It is better written, ‘……….*Campylobacter *species identification.’ Similarly, the binomial nomenclature should be followed throughout the document.Lines 103&104: ‘………pattern of *Campylobacter jejuni* and C. coli among under-five children with diarrhea.’ It is better written as, ‘………pattern of *C. jejuni* and *C. coli* among under-five children with diarrhea.Lines 116, 121, 123, 124, 137 and 128: ‘Bahr Dar’ should be corrected as ‘Bahir Dar’Line 199: ‘Data quality’ better written as ‘Data quality assurance’Lines 232- 235: Table 1 of the last column ‘*Campylobacter *infection (%, CI)’, the confidence interval (CI) was not much important and better be removed. Similar cenario existed in Tables 2 &3.Line 243: ……Campylobacter species, better written as ……..*Campylobacter* species,Line 259: The subheading ‘Potential risk factors and their relationship with Campylobacter enteritis’ It is better written as, ‘Potential associated factors and their relationship with *Campylobacter *enteritis’Line 272: Table 5 should be supplemented with P- value of COR and P-value of AOR.In the result part, you said nothing about how may culture positives confirmed by MALDI-TOF MS assay as positive and how many also became negative.The presentation of associated factors better be presented as, for example lines 276-278: ‘Consequently, children residing in peri-urban areas and having contact with domestic animals were 4.91(1.92-12.54) and 5.03(2.18-11.62) times more likely to contract *Campylobacter *gastroenteritis.’ It is better presented as, ‘Consequently, children residing in peri-urban areas and having contact with domestic animals were 4.91 (95%CI: 1.92-12.54, P=?) and 5.03 (95%CI: 2.18-11.62, P=?) times more likely to contract *Campylobacter *gastroenteritis.’ Sometimes we can also write as (AOR: ?, 95%CI: ? P=?). This works for all associated factors analysis.Lines 425-434: Authors contribution should be removed as the system creates automatically.Beyond the comparison, the discussion part requires scientific explanation and reasoning.It is better also you follow the manuscript writing protocol for PloS One, especially font size, font type, reference list writing, table and figure preparation and whether figures submitted within the main manuscript or not.Proof readig of the whole manucript is needed.

Reviewers' comments:

Reviewer's Responses to Questions

**Comments to the Author**

1. If the authors have adequately addressed your comments raised in a previous round of review and you feel that this manuscript is now acceptable for publication, you may indicate that here to bypass the “Comments to the Author” section, enter your conflict of interest statement in the “Confidential to Editor” section, and submit your "Accept" recommendation.

Reviewer #2: All comments have been addressed

Reviewer #3: (No Response)

2. Is the manuscript technically sound, and do the data support the conclusions?

Reviewer #2: Yes

Reviewer #3: Partly

3. Has the statistical analysis been performed appropriately and rigorously? 

Reviewer #2: I Don't Know

Reviewer #3: Yes

4. Have the authors made all data underlying the findings in their manuscript fully available?

Reviewer #2: Yes

Reviewer #3: Yes

5. Is the manuscript presented in an intelligible fashion and written in standard English?

Reviewer #2: No

Reviewer #3: No

6. Review Comments to the Author

Reviewer #2: I have the following observations:

1. Two versions of the manuscript are presented which are very different. The first version is marked as "revised" and the second marked "Revised manuscript with changes with yellow". I assume the second one was supposed to be the revised manuscript highligting the changes that have been made. The authors need to thoroughly review their work before submission to avoid this type of confusion.

2. There are some grammatical or typographical errors in the manuscript. The article could benefit from thorough language editing. Below are some few examples:

- title - "antimicrobial resistance pattern" an "s" should be added to "pattern"

- Abstract - "children with under-five year age disproportionally affected with foodborne illness."

- "The prevalence of C. jejuni and C. coli among under-five children was 5.1% (CI: 3.0- 7.0%) and 1.9% (CI:0.7-3.3%). - "respectively" may be needed at the end of this statement.

- "The rest might be either refuse to participate or unable to get specimen"

- "Clinical characterization of under-five children" might sound better if written as "Clinical characteristics of children aged under-five years"

3. The details of how data was analysed - especially the bivariate and multivariate are lacking in both abstract and main manuscript. What was the outcome of interest? What measures of association were calculated? How were covariates progressed from bivariate to multivariate models? Since the data

4. Sampling of children - in some places the authors refer to "convenient sampling" yet in other places they refer to "consecutive sampling". They need to be consistent.

5. "The rest might be either refuse to participate or unable to get specimen" - the authors only talk of the 428 participants. This statement implies that there were "others" who are not quantified. It is also not clear what proportion of these "others" were excluded based on the two criteria mentioned. Were these "others" significantly different from the included participants?

6. "Data of the current study showed that 8 variables analyzed by Bivariate logistic regression had P value < 0.25 and considered for further analysis using multivariable logistic regression model.The Model was checked by Hosmer-Lemeshow goodness-of-fit." Should be refiened and moved to the data analysis section rather than results.

Reviewer #3: Basic questions to the Author Lanuage, Fragmented sentence, punctuation and similar questions should edited through out the manuscrpit as indicated in the comment.

Sampling and selection of participants should be clearly indicated

7. PLOS authors have the option to publish the peer review history of their article (what does this mean?). If published, this will include your full peer review and any attached files.

Reviewer #2: No

Reviewer #3: No

---

## [Author Response · Author response to Decision Letter 1]

26 Apr 2024

Point to point response to reviewers

Dear editor and reviewers

We express our sincere appreciation to all individuals who have generously dedicated their time and expertise to provide valuable feedback and suggestions for enhancing our manuscript. Conversely, we have embraced nearly all the comments and suggestions, and have diligently incorporated them into our work. The revisions have been visually emphasized with the use of yellow color.

Additional Editor Comments:

• The title, ‘Campylobacter jejuni and Campylobacter coli infection, determinants and antimicrobial resistance pattern among under-five children with diarrhea in Amhara National Regional state, Northwest Ethiopia’. The word ‘resistance’ better be replaced by ‘susceptibility’. Then modify the objective and description of tables accordingly.

Response: Dear Editor, We extend our appreciation for your valuable comment. However, it is important to note that our study was specifically designed to explore the antimicrobial resistance (AMR) of the Campylobacter species, which has gained significant global recognition. Moreover, all the results have been presented in terms of resistance levels. Hence, we firmly assert that the current topic effectively reflects the findings of our study. Thank you for your understanding. 

• The affiliation better which was ‘5Animal Health Institute, Sebeta, Ethiopia’ should be linked to the author (s) or removed.

Response: Thank you for your comment and we accepted it and corrected accordingly. Highlighted with yellow color at L5 and L15

• The email addresses of authors were not needed here in the main manuscript as it appears in the system except the corresponding author.

Response: Thank you! We accepted the comment and removed the email addresses of authors from the title page except the corresponding author 

Lines 39 &40: ‘Suspected colonies were analyzed using matrix-assisted laser desorption ionization-time of flight mass spectrometry to confirm the species.’ It is better written as, ‘The suspected colonies were analyzed using matrix-assisted laser desorption ionization-time of flight mass spectrometry (MALDI-TOF-MS) to confirm the species.’

Response: Thank you for your comment and we accepted it and corrected accordingly. Highlighted with yellow color at L37

• Lines: 41&42: ‘Logistic regression was used to analyze the associated factors.’ It was better written as, ‘Logistic regression was used to analyze the associated factors with P-value <0.05’.

Response: Thank you for your comment and we accepted it and corrected accordingly. Highlighted with yellow color at L40

• Lines 43-45: ‘Among the 428 samples, 30 (7.0%) (CI: 4.5-9.3%) were culture-positive for Campylobacter species. The prevalence of C. jejuni and C. coli among under-five children was 5.1% (CI: 3.0-44 7.0%) and 1.9% (CI: 0.7-3.3%)’. Here the sign (%) from the CI part should be removed because by default it has already known. Similarly, the number of isolates with the total better written together with the percentages. This works for the whole abstract and result part of the paper. For example, for the above better written as, “Among the 428 samples, 30 (7.0%) (CI: 4.5-9.3) were culture-positive for Campylobacter species. The prevalence of C. jejuni and C. coli among under-five children was 22/428 (5.1%) (CI: 3.0-44 7.0) and 8/428 (1.9%) (CI: 0.7-3.3)’.

Response: Thank you for the comment and we removed the percentage sign from CI throughout the manuscript. 

• Line 64: ‘………health infrastructure is prevalent. [1].’ Better written as, ‘………health infrastructure is prevalent [1].’

Response: Thank you for the comment and we omitted unnecessary full stop and highlighted with yellow color at L62 

• Line 88: ‘……… in animal husbandries. [12,13].’ Better written as, ‘,… in animal husbandries [12,13].’

Response: Thank you for the comment and we omitted unnecessary full stop and highlighted with yellow color at L86

• Similarly, line 94: ‘……………..and tetracycline. [14–16].’ Better written as, ‘…….. and tetracycline [14–16].

Response: Thank you for the comment and we omitted unnecessary full stop and highlighted with yellow color at L92

• Line 96 &97: ‘Binary and multivariable logistic analysis was used to identify associated risk factors. A P value less than 0.05 at a 95% confidence interval was statistically significant’. Better written as ‘Bivariable and multivariable logistic regression analysis was used to identify associated risk factors. A P-value < 0.05 at a 95% confidence interval was considered as statistically significant’.

Response: Thank you for the comment and we accepted and corrected at L194 and 195

• In the data analysis part, it is better you add something about the interpretation of ‘MALDI-TOF MS assay. 

• Line 100: ‘MALDI-TOF MS provides a rapid…….’ It is better written as, ‘Matrix-assisted laser desorption ionization-time of flight mass spectrometry provides a rapid…….’ As the start of a sentence will not be acronyms/abbrevations.

Response: Thank you for the comment and we accepted and corrected at L98-99

• Line 100: ‘…….Campylobacter species identification.’ It is better written, ‘……….Campylobacter species identification.’ Similarly, the binomial nomenclature should be followed throughout the document.

Response: Thank you for the comment and we accepted and italicized at L98

• Lines 103&104: ‘………pattern of Campylobacter jejuni and C. coli among under-five children with diarrhea.’ It is better written as, ‘………pattern of C. jejuni and C. coli among under-five children with diarrhea.

Response: Thank you for the comment and we accepted and italicized at L102

• Lines 116, 121, 123, 124, 137 and 128: ‘Bahr Dar’ should be corrected as ‘Bahir Dar’

Response: Thank you for the comment and we accepted and corrected as Bahir Dar at L115, 120, 122, 123, and 127

• Line 199: ‘Data quality’ better written as ‘Data quality assurance’

Response: Thank you for the comment and we accepted and corrected as ‘Data quality assurance’ at L198

• Lines 232- 235: Table 1 of the last column ‘Campylobacter infection (%, CI)’, the confidence interval (CI) was not much important and better be removed. Similar cenario existed in Tables 2 &3.

Response: Thank you for the comment and we accepted and removed all CI from the three tables. For your information the first version of the tables was without CI. But this was happen after comment from one of the first reviewers. 

• Line 243: ……Campylobacter species, better written as ……..Campylobacter species,

Response: Thank you for the comment and we accepted and italicized as Campylobacter species at L259

• Line 259: The subheading ‘Potential risk factors and their relationship with Campylobacter enteritis’ It is better written as, ‘Potential associated factors and their relationship with Campylobacter enteritis’

Response: Thank you for the comment and we accepted and corrected as Potential associated factors at L268

• Line 272: Table 5 should be supplemented with P- value of COR and P-value of AOR.

 Response: Thank you for the comment and we accepted and included P-value for both COR and AOR in table 5

• In the result part, you said nothing about how may culture positives confirmed by MALDI-TOF MS assay as positive and how many also became negative.

Response: Thank you for the comment and we accepted and articulated culture positive confirmation by MALDI-TOF MS assay at L256-257.

• The presentation of associated factors better be presented as, for example lines 276-278: ‘Consequently, children residing in peri-urban areas and having contact with domestic animals were 4.91(1.92-12.54) and 5.03(2.18-11.62) times more likely to contract Campylobacter gastroenteritis.’ It is better presented as, ‘Consequently, children residing in peri-urban areas and having contact with domestic animals were 4.91 (95%CI: 1.92-12.54, P=?) and 5.03 (95%CI: 2.18-11.62, P=?) times more likely to contract Campylobacter gastroenteritis.’ Sometimes we can also write as (AOR: ?, 95%CI: ? P=?). This works for all associated factors analysis.

Response: Thank you for the comment and we accepted and corrected accordingly at L278, 279, 280 and 281

• Lines 425-434: Authors contribution should be removed as the system creates automatically.

Response: Thank you, we accepted and removed from the manuscript. 

• Beyond the comparison, the discussion part requires scientific explanation and reasoning.

• It is better also you follow the manuscript writing protocol for PloS One, especially font size, font type, reference list writing, table and figure preparation and whether figures submitted within the main manuscript or not.

• Proof readig of the whole manucript is needed

Reviewer #2:

 I have the following observations:

1. Two versions of the manuscript are presented which are very different. The first version is marked as "revised" and the second marked "Revised manuscript with changes with yellow". I assume the second one was supposed to be the revised manuscript highligting the changes that have been made. The authors need to thoroughly review their work before submission to avoid this type of confusion.

Response: We would like to express our gratitude to the reviewer for bringing this critical matter to our attention, as it has alerted us to exercise caution prior to manuscript submission. The discrepancy occurred due to the oversight of not removing the last section, which included author contributions, data availability, consent for publication and so on. However, it is important to note that the body of the text, specifically the findings before the reference section, remained unchanged.

2. There are some grammatical or typographical errors in the manuscript. The article could benefit from thorough language editing. Below are some few examples:

- title - "antimicrobial resistance pattern" an "s" should be added to "pattern"

Response: Thank you! We accepted and corrected accordingly at L2

- Abstract - "children with under-five year age disproportionally affected with foodborne illness."

Response: Thank you! We accepted and corrected accordingly at L27

- "The prevalence of C. jejuni and C. coli among under-five children was 5.1% (CI: 3.0- 7.0%) and 1.9% (CI: 0.7-3.3%). - "respectively" may be needed at the end of this statement.

Response: Thank you! We accepted and corrected accordingly at L43

- "The rest might be either refuse to participate or unable to get specimen"

Response: Thank you! We accepted and omitted the letter‘s’ from the word ‘specimens’ at L224

- "Clinical characterization of under-five children" might sound better if written as "Clinical characteristics of children aged under-five years"

Response: Thank you! We accepted and corrected accordingly at L236

3. The details of how data was analysed - especially the bivariate and multivariate are lacking in both abstract and main manuscript. What was the outcome of interest? What measures of association were calculated? How were covariates progressed from bivariate to multivariate models? Since the data

Response: We value your constructive feedback in order to improve the overall quality of our manuscript. Could you kindly review the detailed concept presented in the 'Data analysis' section, specifically lines 195-204? Furthermore, we have also made revisions to the abstract portion.L39-43

4. Sampling of children - in some places the authors refer to "convenient sampling" yet in other places they refer to "consecutive sampling". They need to be consistent.

Response: Thank you! We accepted and corrected the paragraph to make consistent. L140-145

5. "The rest might be either refuse to participate or unable to get specimen" - the authors only talk of the 428 participants. This statement implies that there were "others" who are not quantified. It is also not clear what proportion of these "others" were excluded based on the two criteria mentioned. Were these "others" significantly different from the included participants?

Response: Your comment and suggestion is greatly appreciated, dear reviewer. The idea has been clarified at L232-234 for better understanding.

6. "Data of the current study showed that 8 variables analyzed by Bivariate logistic regression had P value < 0.25 and considered for further analysis using multivariable logistic regression model.The Model was checked by Hosmer-Lemeshow goodness-of-fit." Should be refiened and moved to the data analysis section rather than results.

Response: Thank you! We accepted and moved to ‘data analysis section’ at L205-206

Reviewer #3: 

• Basic questions to the Author Language, Fragmented sentence, punctuation and similar questions should edited throughout the manuscript as indicated in the comment.

Sampling and selection of participants should be clearly indicated

Response: Dear reviewer, we value your feedback, and one of the authors, who is a native English speaker, has made the required edits to prevent any loss of information and enhance the readability of the manuscript.

The title is very important and it is tries to address the current hot topics and the study makes significant contributions to the study area. But it needs some modifications like

Campylobacter jejuni and coli infection, determinants and antimicrobial resistance pattern among under-five children with diarrhea in Amhara National Regional State, Northwest Ethiopia

Campylobacter species infection, determinants and antimicrobial resistance pattern among under-five children with diarrhea in Amhara National Regional State, Northwest Ethiopia

Response: Dear reviewer we appreciate your feedback. Our team highly regards your concern. The existing title has been carefully chosen to accurately reflect our objective. Furthermore, it is worth noting that the majority of prior research conducted in Ethiopia has focused on the genus level. Hence, this title appropriately highlights the distinctions in our study.

• Check affiliation of each Author and the corresponding number.

Response: Thank you for your comment and we accepted it and linked the author with the affiliation. Highlighted with yellow color at L5 and L15

Response: 

Abstract

• The whole manuscript needs language revision from example in the abstract section the first sentence is fragmented should be revised and the term exploration is more of for qualitative study (line 29-32). 

Response: Thank you for your comments and we amended accordingly as stated at L29-31

• In the abstract section there is repeation of the objective both in the background section and the objective section please make or write one section.

Response: Thank you for your comments and we removed from the background section of the paragraph. 

Methods: 

• The study sites were selected using a random sampling technique, while the study subjects were included using a convenient sampling technique. Data were collected using a structured questionnaire section needs revision.

Response: Thank you for your comments and corrected accordingly at L36

Result:

• The prevalence of C. jejuni and C. coli among under-five children was 5.1% (CI: 3.0-7.0%) and 1.9% (CI: 0.7-3.3%) (line #45) add respectively avoid fragmentation of the sentence.

Response: Thank you for your comments and amended accordingly at L47

• The resident avoid the (line #45-46) 

Response: Dear reviewer we greatly appreciate your feedback and offer our sincere apologies for our failure to remove the mentioned finding from gaining prominence as a result.

• One-third of the Campylobacter isolates (33.3%) (Line # 47-48) use consistent expressions of the frequencies either number in words and percentage or number in a numerical value and percentage. 

Response: Thank you for your comments and corrected accordingly at L50, 51

Conclusion: 

• The prevalence of Campylobacter species was relatively low relative to what. However, a high rate of ciprofloxacin and tetracycline-resistance strains was identified relative to what is your reference. Continuous surveillance on antimicrobial resistance and health education personal and environmental hygiene should be implemented in the community. Indicate the responsible bodies and future perspective for the researchers.

Response: Thank you! We accepted your comments and corrected accordingly at L53 and L54

Introduction

• In thi

---

## [Decision Letter · Decision Letter 2]

13 May 2024

Campylobacter jejuni and Campylobacter coli infection, determinants and antimicrobial resistance patterns among under-five children with diarrhea in Amhara National Regional State, Northwest Ethiopia

PONE-D-24-04047R2

Dear Dr. Worku,

We’re pleased to inform you that your manuscript has been judged scientifically suitable for publication and will be formally accepted for publication once it meets all outstanding technical requirements.

Kind regards,

Tebelay Dilnessa, MSc

Academic Editor

PLOS ONE

Additional Editor Comments (optional):

Some proof reading is required.

Reviewers' comments:

Reviewer's Responses to Questions

**Comments to the Author**

1. If the authors have adequately addressed your comments raised in a previous round of review and you feel that this manuscript is now acceptable for publication, you may indicate that here to bypass the “Comments to the Author” section, enter your conflict of interest statement in the “Confidential to Editor” section, and submit your "Accept" recommendation.

Reviewer #2: All comments have been addressed

Reviewer #3: All comments have been addressed

2. Is the manuscript technically sound, and do the data support the conclusions?

Reviewer #2: Yes

Reviewer #3: Yes

3. Has the statistical analysis been performed appropriately and rigorously? 

Reviewer #2: Yes

Reviewer #3: Yes

4. Have the authors made all data underlying the findings in their manuscript fully available?

Reviewer #2: Yes

Reviewer #3: Yes

5. Is the manuscript presented in an intelligible fashion and written in standard English?

Reviewer #2: Yes

Reviewer #3: No

6. Review Comments to the Author

Reviewer #2: The authors have made a good effort at revising the manuscript, which now reads better. I have no further comments.

Reviewer #3: As it it is indicated above the the revision of manuscripit using standard English should be checked again before publication or acceptance of the document through out the whole document.

7. PLOS authors have the option to publish the peer review history of their article (what does this mean?). If published, this will include your full peer review and any attached files.

Reviewer #2: No

Reviewer #3: **Yes: **Demissie Assegu Fenta

---

## [Editor Report · Acceptance letter]

17 May 2024

PONE-D-24-04047R2 

PLOS ONE

Dear Dr. Worku, 

I'm pleased to inform you that your manuscript has been deemed suitable for publication in PLOS ONE. Congratulations! Your manuscript is now being handed over to our production team.

Kind regards, 

on behalf of

Dr. Tebelay Dilnessa 

Academic Editor

PLOS ONE